# Polyethylene-Matrix Composites with Halloysite Nanotubes with Enhanced Physical/Thermal Properties

**DOI:** 10.3390/polym11050787

**Published:** 2019-05-02

**Authors:** Janusz W. Sikora, Ivan Gajdoš, Andrzej Puszka

**Affiliations:** 1Department of Technology and Polymer Processing, Mechanical Engineering Faculty, Lublin University of Technology, 20-618 Lublin, Poland; 2Department of Technologies and Management, Mechanical Engineering Faculty, Technical University of Košice, 040 01 Košice, Slovak Republic; ivan.gajdos@tuke.sk; 3Department of Polymer Chemistry, Faculty of Chemistry, Maria Curie-Sklodowska University, 20-614 Lublin, Poland; andrzej.puszka@umcs.pl

**Keywords:** low-density polyethylene, halloysite nanotubes, mechanical properties, nanomaterials, polymer composites, thermal properties

## Abstract

The aim of the present work is to investigate the effect of halloysite nanotubes (HNT) on the mechanical properties of low-density polyethylene composites modified by maleic anhydride-grafted PE (PE-*graft*-MA). Polyethylene nanocomposites were prepared using an injection molding machine, Arburg Allrounder 320 C 500–170; the HNT content was varied at 0 wt %, 2 wt %, 4 wt % and 6 wt %, and the PE-*graft*-MA content was varied at 5 wt %. The composites were examined for their ultimate tensile stress, strain at ultimate stress, hardness, impact strength, melt flow rate, heat deflection temperature, Vicat softening temperature, crystallinity degree and phase transition temperature. It was found that the addition of halloysite nanotubes to low-density polyethylene (LDPE) led to an increased heat deflection temperature (HDT, up to 47 °C) and ultimate tensile strength (up to 16.00 MPa) while the Vicat softening temperature, strain at ultimate stress, impact strength and hardness of examined specimens slightly decreased. Processing properties of the materials specified by the melt flow rate (MFR) deteriorated almost twice. The results have demonstrated that the nanoparticles can reinforce enhance LDPE at low filler content without any considerable loss of its ductility, but only when halloysite nanotubes are superbly distributed in the polyethylene matrix.

## 1. Introduction

Polymers are now used in almost all sectors of the economy, including automotive, electrical and electronic industries, the production of medical and office equipment and supplies, construction, packaging, the manufacture of gardening equipment, home appliances, consumer electronics and furniture. One group of polymers that account for half of the world’s demand are polyolefins. The market for these materials is so large because they are relatively inexpensive, have multifarious physicochemical properties and good processability and can be produced by many processing methods [1,2].

Particularly popular are polymers filled with mineral nanofillers [3,4] characterized by specific properties [5,6,7]. Even a small content of nanofiller (up to several percent by weight) can positively affect selected properties of polymeric materials [8,9,10,11]. Research has shown that nanoparticles added to olefin plastics can effectively increase their thermal stability and fire resistance [12,13,14], as well as reduce their thermal expansion coefficient [15,16]. In many cases, the mechanical and processing properties of materials with nanofillers do not deteriorate, and, in some cases, they improve. Most studies reported in the literature reveal that the impact strength and elongation at break of such materials decrease when a nanofiller is added or when its content is increased [17,18].

In study [19], the fire resistance of extruded polyamide 6 significantly increased after the addition of over 15 wt % halloysite nanotubes (HNT). A similar observation was made during thermal stability and flame retardancy tests of low-density polyethylene composites with the addition of HNT varied from 2 wt % to 20 wt % [20]. The tensile strength and elongation of this material were also slightly higher than those of low-density polyethylene (LDPE). Compared to LDPE, HNT composites with a low-density polyethylene matrix showed increased tensile strength and Young’s modulus; they also had higher thermal stability [21]. The next paper [22] describes the tests and results only of mechanical properties of LDPE filled with HNT samples obtained in the injection process. Mechanical properties of the LDPE (Young’s modulus, tensile stress, strain at strength, impact strength and hardness) were changed be adding a nanofiller. With increasing content of the nanofiller strain, the strength and impact strength decreased (from 80.27% to 68.10% and from 16.2 kJ/m^2^ to 14.66 kJ/m^2^, respectively).

In epoxy resins, the addition of 2.3 wt % HNT was observed to cause a fourfold increase in their stiffness, impact strength, tensile strength and thermal resistance [23]. In study [24], LDPE with 5 wt % HNT had lower mass flow rate index, tensile strength and elongation at break, but the impact strength of the test pieces increased. Similar effects of the same amount of HNT on the above-mentioned properties were observed in tests involving PP extrusion [24]. In study [25], the addition of 2 wt % HNT to HDPE resulted in decreasing the melting point of the obtained composite with simultaneous reduction in the size of crystallites, their size distribution and crystallinity degree [25]. In paper [26], it was shown that the addition of 8 wt % HNT to LLDPE led to an increase in the crystallization temperature, crystallization degree and tensile modulus, while causing a decrease in tensile stress at break and elongation at break of the polymer. During the tests of UHMWPE filled with up to 8 wt % HNT, the total property of the UHMWPE fiber generally improved with the addition of HNTs in specific concentrations [27]. In another work [28], the occurrence of aggregate defects formed by high concentration of HNT causes some damage to the crystallinity and mechanical properties of UHMWPE nanocomposites.

As these examples show, evidence reported in the literature is inconclusive, indicating that the impact of nanofillers and their concentrations on the properties of obtained nanocomposites must be tested individually for every matrix and nanophase combination.

This work aims to determine the effect of adding HNT on selected mechanical, processing and thermal properties of LDPE, a polymer that can easily be processed by innovative extrusion methods [29,30] and is thus widely used in the manufacture of electric wires, cables, films, pipes and containers [31,32].

## 2. Materials and Methods

### 2.1. Materials

Experimental tests were performed on low-density polyethylene (LDPE), marketed under the trade name Malen E and symbol FGAN 18-D003, produced by Basell Orlen Polyolefins (Plock, Poland). According to the data provided by the producer, this material exhibits the following properties: Density = 921 kg/m^3^, melt flow rate (190 °C; 2.16 kg) = 0.28 g/10 min, tensile strength at break = 21 MPa, tensile strain at break = 360%, tensile modulus = 220 MPa, Vicat softening temperature = 93 °C and shore hardness D = 50°. The LDPE used in the experiments is suitable for extrusion applications. The recommended processing temperature for this product ranges from 170 °C to 220 °C.

Halloysite nanotubes (HNTs) were manufactured by Sigma-Aldrich (Milwaukee, WI, USA) and delivered in powder form. The size of HNT particles varied within 30–70 nm of diameter and 1–3 µm of length; they had a specific surface area of 64 m^2^/g and a density of 2530 kg/m^3^. Halloysite is a natural nanosized tubular clay mineral that has many potentially important applications in different industrial fields [33].

Polyethylene grafted with maleic anhydride (PE-*graft*-MA), available from Sigma-Aldrich (Milwaukee, WI, USA), was used as a compatibilizer. It has a melting temperature of 105 °C and a density a 920 kg/m^2^.

### 2.2. Sample Preparation

The samples for studies of the melt flow rate (MFR), phase transition temperature, crystallinity degree and heat deflection temperature (HDT), Vicat softening temperature (VST), as well as samples for studies of ultimate tensile stress, strain at ultimate stress, hardness and impact strength were obtained using an injection molding machine, Arburg Allrounder 320 C 500–170 (ARBURG GmbH, Lossburg, Germany). An injection molding machine enables injection up to 80 cm^3^ of polymer plasticized with the maximum injection pressure up to 200 MPa into the mold cavity. This machine had two mold cavities, their shape and size conforming to those of standardized samples for strength testing.

The temperature of the heating zones of the injection molding machine’s plasticizing system, starting with the hopper, was set as 100 °C, 140 °C, 160 °C, 170 °C and 180 °C, while the temperature in the feed opening zone was set at 30 °C. The total time of the injection process cycle was equal to 34.16 s, including the cooling time of 20 s. The composite polymer was injected into the mold at a pressure of 100 MPa and the holding pressure was set to 85 MPa. The temperature of the injection mold was set at 18 °C. Samples were prepared as shown in Table 1.

### 2.3. Methods

*Mechanical properties*. The static tensile test was prepared in accordance with the EN ISO standard 527:2012. Tensile test samples were analyzed using the TIRAtest 2300 (TIRA GmbH, Schalkau, Germany) tensile testing machine at a transducer load force of 10 kN and a testing speed of 100 mm/min. Hardness was determined according to the hardness measurement procedure described in the EN ISO 868:2005 using a conventional testing machine apparatus and a load of 5 kg, while impact strength was determined by the Charpy method in accordance with the EN ISO standard 179-1:2010 using a 7.5 Joule Charpy impact hammer, PSW60/500 (Zwick Roell, Ulm, Germany). As the result of measurement, the average value of the measurement was 10 samples.

*Thermal properties*. Melt flow rate (MFR) was measured in accordance with the ISO standard 1133-1:2011, at 190 °C and a load of 2.16 kg using the Zwick 4105.100 (Zwick Roell, Ulm, Germany). An extrusion plastometer with additional equipment and the PRL TA14 analytical scale. As the result of measurement, the average value of the measurement was 10 samples.

The heat deflection temperature (HDT) test was performed according to ISO 75:2013, with the load exerting a constant bending stress of 0.45 MPa. As the result of measurement, the average value of the measurement was 10 samples.

The Vicat softening temperature (VST) test was performed according to ISO 306. The VST was measured by the A50 method, in which a sample was loaded with a force of 10 N and the temperature increase rate was 50 °C/h. Both the HDT and VST tests were performed using a CEAST device equipped with three workstations (CEAST, Pianezza, Italy). As the result of measurement, the average value of the measurement was 10 samples.

A differential scanning calorimeter DSC 204 F1 Phoenix (NETZSCH, Günzbung, Germany) equipped with the NETZSCH *Proteus* thermal analysis software was used to determine the temperature of phase transitions and the degree of crystallinity. To determine the melting and crystallization temperatures (*T_m_* and *T_c_*, respectively), the tests were carried out in the following cycle: Heating (I) from −100 °C to 150 °C, cooling from 150 °C to −100 °C and reheating (II) to 150 °C. The glass transition temperature (*T_g_*) was determined in an additional heating cycle ranging from −150 °C to 170 °C. Heat was supplied to the pans at a rate of 10 K/min, and argon (purge gas) was admitted therein. Degree of crystallinity (*X_c_*) values for varying HNT content in LDPE were then calculated by thermograms using the equation:$$X_{c}=\frac{\Delta H_{m}}{\left(1-w_{t}\right)\Delta H_{m}^{0}}\times 100\%$$
where $\Delta H_{m}^{0}$ stands for the theoretical specific melting heat of 100% crystalline sample, and *w_t_* represents weight fraction of added HNTs. For LDPE samples, $\Delta H_{m}^{0}$ value was taken as 293 J/g [34].

## 3. Results and Discussion

### 3.1. Morphology of HNT and LDPE Nanocomposites

First, the dispersion of HNT in the polyethylene matrix will be presented. FEI’s scanning electron microscope Nova NanoSEM 450 was used for microscopic examination of the material structure. Figure 1a–d show the results of microscopic examination (SEM) of the polymer composites containing halloysite nanotubes and a compatibilizer. Figure 1a shows the surface of pure polyethylene without additives. The surface was homogeneous and there were no visible inclusions, even at a large magnification.

It has been found that, irrespective of the HNT content, obtained PE/HNT composites were neither perfect nor homogenous in terms of HNT particles distribution. Figure 1b–d point to a poor distribution of HNT particles within the PE matrix at different HNT contents. One can clearly notice the agglomeration of nanotubes and their poor dispersion. This poor distribution can be attributed to the fact that HNT is hydrophilic by nature whereas PE has hydrophobic properties, which is responsible for the weak interaction and compatibility [8,26]. Nanotube agglomerates of various sizes (from a few to several dozen μm) are visible on the polymer surface. Given the size of the nanotubes themselves, the size of the agglomerates was quite large. The polymer did not adhere smoothly to their surface, which may indicate poor adhesion of the agglomerates to the polymer. The higher the HNT content was, the larger and the more densely arranged the agglomerates became. A large concentration of nanotube agglomerates can be observed in Figure 1d. The presence of such structures, also reported in [29], had a negative effect on properties of the entire composite material.

### 3.2. Thermal Properties of LDPE/HNT Nanocomposites

An analysis of the thermal images obtained from the first and second heating of the samples as well as the data in Table 2 demonstrate that the melting point and the glass transition temperature of obtained composites were slightly lower than the corresponding LDPE temperatures.

However, the crystallization temperature of the composites was higher than that of the unmodified LDPE, which indicates that the crystallites in the composites were smaller. The observed lower melting peak widths of the composites prove that the size distribution of LDPE crystallites decrease with the addition of the nanofiller.

The degree of crystallinity of produced composites (obtained for the first heating scan) first increased and then decreased. The increase in the degree of crystallinity at 2% nanofiller content resulted from the fact that the nanofiller acts as a nucleating agent promoting the formation of crystallization nuclei. Results obtained from the second heating scan were inconsistent. Generally, the *T_m_* values decreased with increasing nanofiller content. Additionally, decreasing melting temperatures may be related to the presence of PE-*graft*-MA, which *T_m_* values were 105 °C whereas *T_m_* of LDPE was 116.2 °C. Moreover, the *T_m_* values from the first heating scan were higher than those from the second heating scan. This may be due to a better rearrangement and more perfect form and/or thicker crystals. The nucleating effect of HNT has already been reported with respect to PA6 [5] and PP nanocomposites [35]. This effect is usually associated with a reduced size of polycrystalline aggregates and increased content of the crystalline phase. A subsequent decrease in the degree of crystallinity was, in turn, probably related to the fact that crystallites grow more slowly due to reduced mobility of individual macromolecule segments in the nanoparticle environment. The enthalpy of the tested composites was lower than that of LDPE. With increasing HNT content, the glass transition temperature slightly decreased, but this decrease did not seem to directly depend on nanofiller content.

Besides glass transition regions and melting regions, DSC curves from the first heating (Figure 2) also exhibited a third transition appearing around 50 °C. According to the authors at work [36], it could be due to variety stress, probably molecular orientation, frozen during processing and which recovers during DSC heating or it has also been proposed that the apparition of several peaks represent the melting of lamellae of different size. The local melting of the smallest sizes may occur before the one of the thickest sizes [37].

The relationship between the heat deflection temperature (HDT) and nanofiller mass content with and without 5 wt % compatibilizer is shown in Figure 3.

The tests have demonstrated that the addition of a nanofiller in the form of halloysite nanotubes had a positive effect on the heat deflection temperature, causing it to increase slightly. The addition of a 2 wt % nanofiller to low-density polyethylene made the HDT increase by almost 1 °C, while on the addition of 6 wt % HNT the HDT increased by almost 3.5 °C Although insignificant, such changes can be explained by the fact that polymers with a larger degree of crystallinity would generally have a higher melting point. Materials with higher melting points are mechanically more rigid (including the HDT) than those characterized by lower melting points [38], which agrees with the obtained results (the Young’s modulus increased while the strain at break decreased).

The dependences between the Vicat softening temperature (VST; Figure 4) and nanofiller mass content in the polyethylene matrix with a 5 wt % compatibilizer is presented in Figure 4.

The addition of a nanofiller in the form of halloysite nanotubes into low-density polyethylene caused a slight decrease in the Vicat softening temperature. After the addition of a 2% nanofiller the VST decreased by almost 1 °C, while the addition of a 6% nanofiller made the VST drop by 2 degrees. The Vicat softening temperature was identified with the glass transition temperature, and its decrease (Table 2) accounted for the decrease in the Vicat point value.

The tests also involved the determination of the melt flow rate of produced nanocomposites. It could be observed that the melt flow rate of the obtained polymer nanocomposites increased (Figure 5). The melt flow rate of the composite with 2 wt % nanofiller content of halloysite nanotubes increased in comparison with the virgin polymer from 0.318 g/10 min to 0.488 g/10 min, which amounted to an increase by 53.45%. With increasing nanofiller content, the MFR increased to 0.608 g/10 min, which amounts to more than 24%. One could also observe a slight increase in the nanocomposite processability with increasing HNT content when filling high-density polyethylene and polypropylene with this filler [25]. The addition of HNT affects the flow behavior of the polymer, which, however, can still be processed as a pure material.

### 3.3. Mechanical Properties of LDPE/HNT Nanocomposites

Strength results of obtained polyethylene nanocomposites are given in Table 3.

The results demonstrate that the highest Young’s modulus of 167 MPa was exhibited by pure LDPE while the lowest Young’s modulus (146 MPa) was obtained for LDPE with 2% HNT and 5% PE-*graft*-MA. The Young’s modulus decreased after the addition of the filler into the basic material. A somewhat different course of this relationship was presented in [22], in which a slight increase in Young’s modulus was observed. However, these studies have been presented without statistical analysis, and the biggest difference in the results did not exceed 3%.

The largest reduction in the Young’s modulus of 21 MPa was obtained for the LDPE/2% HNT/5% PE-*graft*-MA material. This reduction in the Young’s modulus amounted to 12.6% compared to the Young’s modulus determined for the LDPE material without the filler.

The reinforcing effect produced by the addition of the nanofiller was most often explained by the formation of an interphase region between the matrix and the particles. Studies have also shown that nanoparticles agglomeration was one of the significant mechanisms leading to increased stiffness of polymer nanocomposites [39]. A recent approach took account of the stiffening effect produced by nanoparticles composed of primary aggregates and agglomerates [40].

Results showed that pure LDPE exhibited the highest strain at UTS (80.27%) while the lowest strain at UTS (68.10%) was obtained for the LDPE material with 6% HNT and 5% PE-*graft*-MA. The nature of these changes was also confirmed in the paper [22]. The tensile strength at break and strain at UTS decreased with increasing filler content in LDPE/HNT nanocomposites, probably because of nanofiller aggregation [19]. The presence of HNT particles led to a stress concentration. Furthermore, the interfacial failure might be more pronounced with increasing halloysite loading.

The results demonstrated that the strain at UTS was the same as the strain at break.

Based on the tensile stress results, one can observe that adding HNT into pure LDPE caused an increased ultimate tensile strength up to 16.00 MPa, which is consistent with the results presented in the work [22]. This was most likely due to filler agglomeration and poor filler interfacial interaction. The difference between the measured values was minimal.

The above results showed that the ultimate tensile strength was the same as the tensile strength at break.

The results of impact strength obtained for the tested samples are given in Figure 6.

Based on the Charpy impact test results one could observe that the addition of fillers and compatibilizer led to a decrease in the impact strength of the tested material. The highest impact strength of 17.20 kJ/m^2^ was exhibited by pure LDPE while the lowest impact strength (14.66 kJ/m^2^) was obtained for the material with 6% HNT and 5% PE-*graft*-MA.

Like before, this could be explained by the poor HNT dispersion that leads to lower interfacial interaction [29]. It can be seen that the addition of halloysite nanotubes reduced the impact strength of produced nanocomposites. This impact strength decrease resulted from the fact that increasing the nanotube content leads to the formation of aggregates, causing a stress concentration in the sample, which initiates brittle fracture. Even the addition of the compatibilizer did not enhance the interfacial interaction between the filler and matrix, leading to a further reduction in the impact strength of the matrix, since the impact strength of LDPE/HNT/PE-*graft*-MA was lower than that of pure LDPE. Similar observations were made in the tests investigating high-density polyethylene filled with HNT [22,41]. A partial break was observed for all samples.

Hardness results of the tested samples are given in Figure 7. According to the results, the highest hardness of 52.3 °ShD was obtained for pure LDPE, while the material with 4% HNT and 5% PE-*graft*-MA exhibited the lowest hardness (49.3 °Sh D), as other researchers have also noticed, e.g., in work [22].

Such a relationship between hardness and nanofiller content is roughly in accordance with the data obtained for Young’s modulus.

## 4. Conclusions

A survey of the state of the art shows that there are more and more studies investigating the influence of halloysite nanotube content on selected properties of polyolefin polymers, and there is a growing interest in HNT as a natural nanofiller for polymers. HNTs are promising nanofillers that can improve thermal stability and flame retardancy of polyolefins and their blends. This study has demonstrated that halloysite nanotubes are poorly distributed in the polyethylene matrix and tend to agglomerate, and therefore the standard plasticizing system is not capable of mixing them properly. It has been shown that the addition of the PE-*graft*-MA compatibilizer did not significantly improve the distribution of halloysite nanotubes in the polyethylene matrix. Results of the tensile, hardness and impact tests have demonstrated that the nanoparticles could reinforce LDPE at low filler content, without any considerable loss of its ductility.

The observed slight changes in the HDT and VST alongside the melt flow rate increase generally proved to be positive features of this nanofiller. To achieve a better distribution of nanotubes in a semi-crystalline and nonpolar matrix, it is recommended to perform a surface treatment.

## Figures and Tables

**Figure 1 polymers-11-00787-f001:**
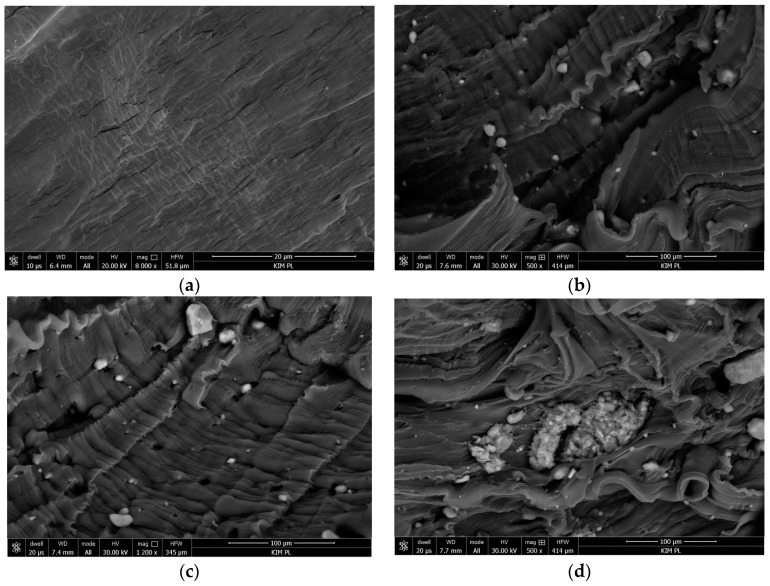
Images showing polyethylene without and with additives: (**a**) Sample No. 1, (**b**) sample No. 2, (**c**) sample No. 3 and (**d**) sample No. 4.

**Figure 2 polymers-11-00787-f002:**
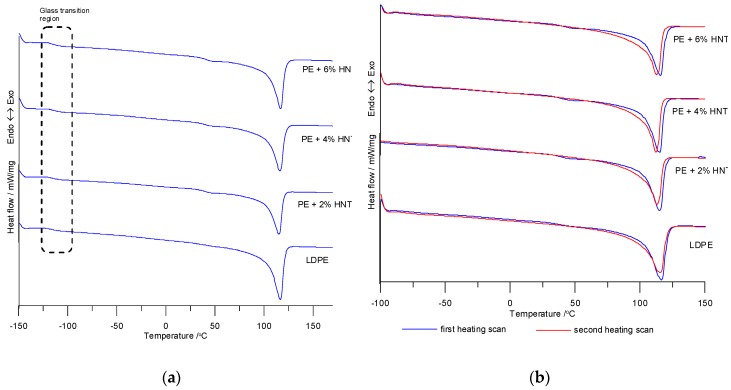
DSC curves from heating: (**a**) Heating from −150 °C to 170 °C and (**b**) two heating scans from −100 °C to 150 °C.

**Figure 3 polymers-11-00787-f003:**
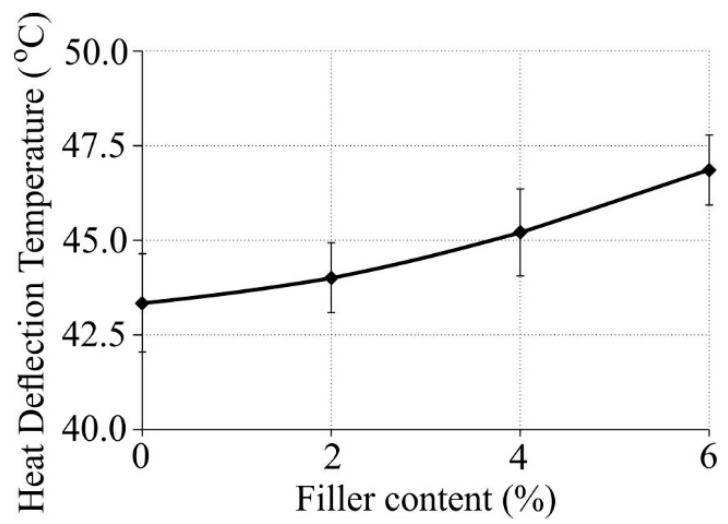
Heat deflection temperature (HDT) versus nanofiller mass content in the composite.

**Figure 4 polymers-11-00787-f004:**
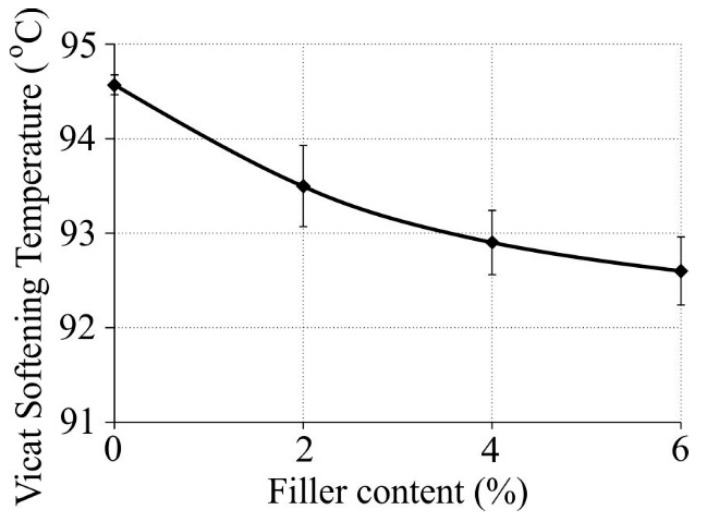
Vicat softening temperature (VST) versus nanofiller mass content in the composite.

**Figure 5 polymers-11-00787-f005:**
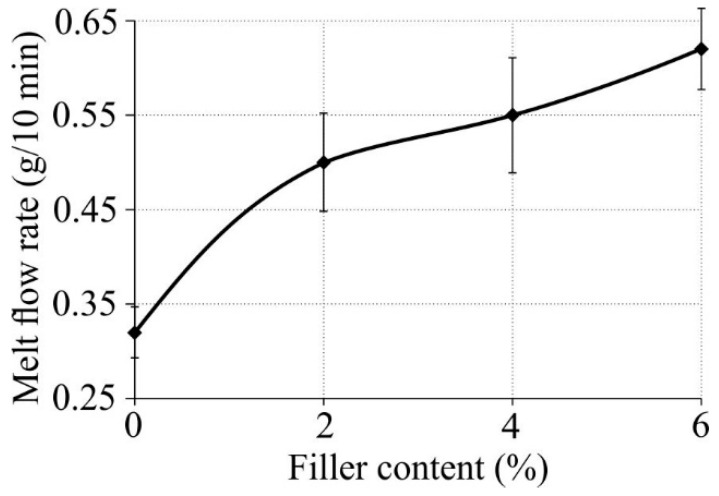
Melt flow rate (MFR) of produced nanocomposites versus nanofiller mass content.

**Figure 6 polymers-11-00787-f006:**
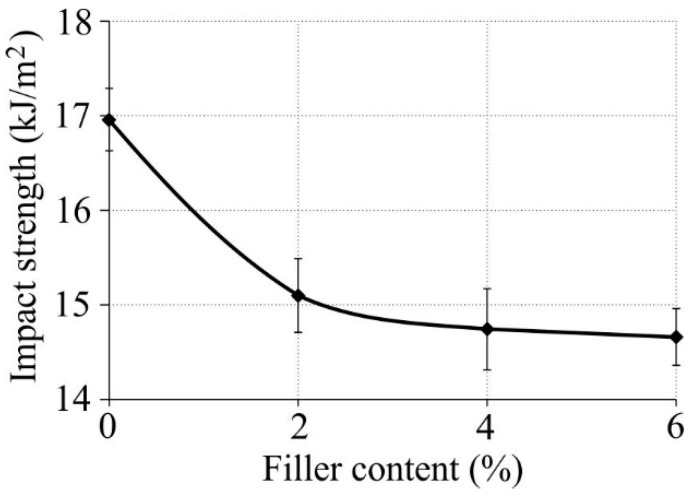
Impact strength of produced nanocomposites versus nanofiller mass content.

**Figure 7 polymers-11-00787-f007:**
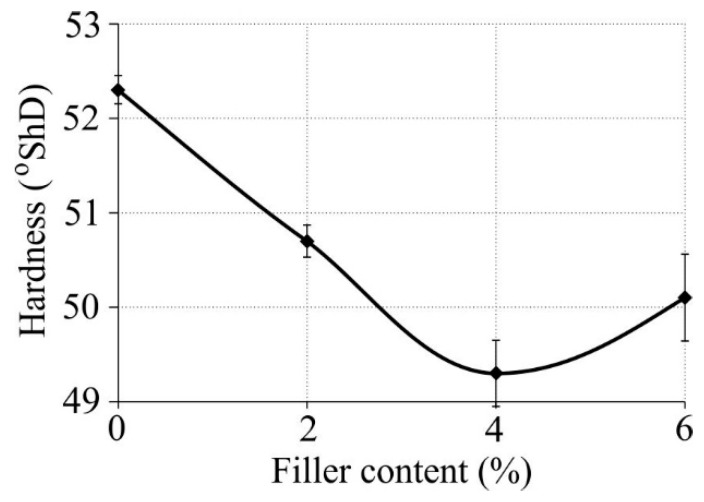
Hardness of produced nanocomposites versus nanofiller mass content.

**Table 1 polymers-11-00787-t001:** Composition of PE/halloysite nanotube (HNT) nanocomposites.

Sample	PE wt %	HNT wt %	PE-*graft*-MA
1	100	0	0
2	93	2	5
3	91	4	
4	89	6	

**Table 2 polymers-11-00787-t002:** Glass transition temperature, melting and crystallization characteristics of low-density polyethylene (LDPE) and relative nanocomposites.

Sample	Heating I				Cooling		Heating II		
	*T_m_* (°C)	Δ*H_m_* (J/g)	*X_c_* (%)	*T_g_* (°C)	*T_c_* (°C)	Δ*H_c_* (J/g)	*T_m_* (°C)	Δ*H_m_* (J/g)	*X_c_* (%)
1	116.2	163.3	55.73	−114.8	92.5	155.3	115.4	163.4	55.77
2	115.0	161.0	56.07	−113.5	94.6	157.9	113.2	154.6	53.84
3	115.0	155.3	55.21	−114.7	95.0	154.9	112.2	154.5	54.93
4	115.5	154.2	55.99	−113.4	94.0	149.2	112.8	163.8	55.47

*T_m_*—melting temperature; Δ*H_m_*—melting enthalpy; *X_c_*—degree of crystallinity; *T_c_*—crystallization peak temperature; Δ*H_c_*—crystallization enthalpy; *T_g_*—glass transition temperature.

**Table 3 polymers-11-00787-t003:** Strength characteristics of LDPE and relative nanocomposites.

Sample	Young’s Modulus (MPa)	Ultimate Tensile Strength (MPa)	Tensile Strength at Break (MPa)	Strain at UTS (%)	Strain at Break (%)
1	167 ± 8	14.68 ± 0.6	14.68 ± 0.6	89.27 ± 2.7	89.27 ± 2.7
2	146 ± 14	15.84 ± 0.8	15.84 ± 0.8	75.19 ± 3.2	75.19 ± 3.2
3	154 ± 16	15.79 ± 1.0	15.79 ± 1.0	72.75 ± 4.3	72.75 ± 4.3
4	162 ± 13	16.00 ± 0.9	15.00 ± 0.9	68.10 ± 3.7	68.10 ± 3.7

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
