# Peer review of "Polyethylene-Matrix Composites with Halloysite Nanotubes with Enhanced Physical/Thermal Properties"

_polymers, 2019, doi:10.3390/polym11050787_

Round 1
Reviewer 1 Report
Polyethylene-Matrix Composites with Halloysite Nanotubes
In this work polyethylene composite materials filled with halloysite were studied. To improve the compatibility maleic anhydride grafted polyethylene (PE-graft-MA) was used. Mechanical and thermal properties were evaluated with the aid of conventional characterization techniques. I found this research work very interesting. However, I have some comments and suggestions in relation to the document. Please find below my main comments.
Abstract:
Please kindly define the terms “HDT” and “MFR”, as it is the first time that it is mentioned in the text.
1. Introduction:
The state of the art and the introduction could be improved. Please kindly check the work entitled “The Effect of Adding Halloysite Nanotubes as Filler on the Mechanical Properties of Low-Density Polyethylene” Materials Science Forum ISSN: 1662-9752, Vol. 919, pp 144-151. (2018). In my opinion, this reference should be cited. Also, the novelty of this work should be highlighted emphasizing indicating the differences with former studies.
2. Materials and methods
Please kindly highlight difference with former studies. More specifically, in relation to the experimental part, exactly the same composite materials were prepared (0ª; 2; 4 and 6%) aand exactly the same amount of PE-graft-MA. Also, the suppliers are the same in both cases.
In section 2.2. please indicate the term “MFR”
In section 2.3. the temperature before table 1, should it be 180 ºC?
3.2. thermal properties of LDPE/HNT
Regarding Table 2, it is observed that as the amount of particles increases, the Tm decreases compared to the neat polymer. The interpretation could be improved. The fact that Tm decrease with the content in particles may be related to the presence of PE-grafted-MA which Tm = 105 ºC whereas Tm of LDPE is 115-116ºC.
I recommend showing DSC traces to observe the transitions.
3.3. Mechanical properties.
I recommend discussing this part considering previously results published in relation to this system (Materials Sci Forum, 2018) and emphasizing the novelty of this work.
Based on my comments, my general recommendation is to write a new version of the document highlighting the novelty of the work and the main differences with previously published material.
Author Response
Dear Reviewer,
I would like to thank very much the Reviewer for valuable remarks and observations that will undoubtedly raise the scientific and cognitive value of the manuscript. Our corrections in the manuscript have been printed in red.
1) According to the reviewer's suggestions, the terms "HDT" and "MFR" have been defined in the summary.
2) The state of the art and introduction have been improved. Several new literature items have been included, including the publication in the Materials Science Forum, which was suggested by the Reviewer.
3) The reviewer is right that the same composite materials were prepared for the experimental part, in the same quantity, type and from the same suppliers, but it should be noted that the research is significantly extended and includes studies of thermal and processing properties, including HDT, MFR, VST, morphology of received materials and tests using DSC.
4) Section 2.2 defines the term "MFR", while in Section 2.3. the temperature of the injection mould is correct and amounted to 18 OC.
5) Of course, the interpretation has been improved. An appropriate explanation has been added to the text. The text also includes additional DSC charts showing DSC traces to observe the transitions.
6) In section 3.3. reference was made to the results in the indicated publication, published in the Materials Science Forum. When analysing the results, a small mistake was found in Table 3, which was corrected.
Once again, I would like to thank the Reviewer for the time he devoted to reviewing the manuscript and for valuable observations.
Best regards,
Authors
Reviewer 2 Report
The prepared manuscript deals with the fabrication of Polyethylene/HNT nanocomposites with some mechanical properties. Detailed results on SEM, DSC, HDT, and VST characteristic were reported. This manuscript needs some revision before consideration to be published in the journal.
1. Some property achievement of the nanocomposites need to mentioned in the current title of the manuscript. For instance “Polyethylene-Matrix Composites with Halloysite Nanotubes with enhanced physical/thermal properties.”
2. Results revealed in Abstract should be more quantitative with some comparison. The optimum amount of HNT should be mentioned in the Abstract.
3. In material section, the authors might need to mention about molecular weight of PE, and how much maleic anhydride percentage in HDPE-g-MA. Additionally, need to explain the structure about HNT through SEM or TEM.
4. How the authors calculated crystallinity for the composites? Using any formula? If so, need to mention in the manuscript. If possible the authors also need to provide DSC plots in revised manuscript.
5. The references inserted in introduction and result discussion part should be updated by 2019. Also, needs rewrite with comparison with previous published results.
6. Some recent publications are needed to cite in the manuscript: Composites Part A: Applied Science and Manufacturing 114, 30-39.
7. Digital images of synthesized materials should insert with proper schematic diagram. Possible to check following paper and cite it.
8. The used instruments needs its country details regarding its perchaing. Check it and modify the manuscript.
Author Response
Dear Reviewer,
I would like to thank very much the Reviewer for valuable remarks and observations that will undoubtedly raise the scientific and cognitive value of the manuscript. Our corrections in the manuscript have been printed in red.
1) According to the reviewer's suggestion, the title has been changed and now it corresponds better to the content.
2) The abstract, which discloses the results in a quantitative context and refers to optimal HNT content, has been corrected.
3) In our opinion, the explanation of the structure of HNT through SEM or TEM is not necessary in this work. Such structures obtained by SEM are generally available. We do not know the molecular weight of the PE and what maleic anhydride percentage in LDPE-g-MA is, but it seems that they are not necessary data needed to understand the observed phenomena. This is the information you need to characterize these materials in detail.
4) According to the suggestion of the Reviewer, a formula has been given in the work, according to which the crystallinity of the obtained composites was calculated. Several DSC plots have also been given, showing DSC curves to observe the transitions.
5) and 6) In the introduction and result discussion, a few recent publications have been introduced, among others those proposed by the Reviewer. The order of the items of references has been changed.
7) Digital images of synthesized materials are shown in Figure 1a-d, while DSC diagrams have been added in the appropriate place of the manuscript.
8) For each instrument used for research, details of the country of origin have been provided.
Once again, I would like to thank the Reviewer for the time devoted to reviewing the article and for valuable observations.
Best regards,
Authors
Reviewer 3 Report
In this work, the authors have investigated the thermal and mechanical properties of low-density polyethylene (LDPE)/halloysite nanocomposites, compatibilized with PE-graft-maleic anhydride. Unfortunately, the work is not suitable for publication in Polymers. Novelty of the work is not obvious in the manuscript, and the experimental approach, methodology, analysis of data, discussions, and conclusions are flawed. In what follows, my concerns are listed:
1. The authors fail to justify the novelty of their work and its significance compared to numerous similar works published in literature. The Introduction section is very general, redundant, and non-specific, and, therefore, not appropriate for the polymer community. The authors need to rewrite the introduction section and focus more on RELEVANT prior work and how their work is building upon previous knowledge. They further need to focus on a specific application where their proposed nanocomposite formulation can provide benefit to the final product.
2. It would be helpful to the reader to just summarize the injection molding process parameters used in this study in a suitable table alongside the material compositions (Table 1), instead of explaining all that information in the text. Most likely, the readers have access to the Arburg Allrounder 320 c manual online!
3. The weight percentage given for PE in Table 1 is awkward, as the total weight percentage does not add up to 100%, given the wt% choices for HNT and PE-graft-MA! Delete the PE wt% column, as it’s understood that the balance is PE.
4. The authors should have made use of statistical design of experiments, specially response surface methodology, for their mechanical and thermal analyses of the specimens (Table 1) followed by optimization. It turns out a properly designed experiment could have revealed useful interaction(s) between the factors, and subsequently made it possible to perform a multi-objective optimization (given the many responses measured in this work). My feeling is that the authors are not familiar with the principles of good experimentation practices that would yield unbiased statistically significant results. This lack of experimental design significantly limits their findings and interpretations of the results.
5. The authors do not mention in the Methods section how many replicates they have for each composition. This information is hidden in the figure captions! Why not mention it once in the Methods section?!
6. The variations in the collected data point values presented as error bars in the figures are most likely related to the different measurements made on the same processed material (different molded specimens, but same injection). This data is not useful for the calculation of variations between the different compositions (treatments). For that to be analyzed, at least a factorial design is needed.
7. Without proper statistical analysis, there is no way to tell whether the glass transition temperatures given in Table 2 are different from each other or not. “Slightly lower” is not appropriate (see Page 4). The same goes for “higher” crystallization temperature of the composites than that of the unmodified LDPE.
8. On Page 5, the authors mention “addition of nanofiller in the form of halloysite nanotubes has a positive effect on the heat deflection temperature.” Then they contradict themselves on the next page and say “Although insignificant, … .” They are right, the effect is insignificant, maybe because the levels they selected for the HNT wt% are too close to each other. In a proper design, they should have investigated what ranges to choose for their factorial levels! The same goes for all the other results presented and discussed.
9. The data presented in Table 3 are mostly overlapping, so it is not obvious to the reader that a certain specimen is superior or inferior to any other one with respect to any specific property of combination of properties. The latter cannot be assessed since the authors did not perform an optimization. There is not a designed experiment, let alone optimization!
Author Response
Dear Reviewer,
I would like to thank very much the Reviewer for valuable remarks and observations.
However, we do not agree with all comments and some of them could be questioned.
1) The introductory part is a bit general, but sticks to the subject, there are many cited publications relating to the impact of filling polymers, mainly PE, with halloisyte nanotubes (HNT), on various properties. In our opinion, it is appropriate for the polymer community, and the opinion of the Reviewer is somewhat subjective. Many conventions of introduction can be chosen and ours is not bad, because it was not questioned by two other reviewers. Similarly to many works, ours does not indicate the specific application of the obtained nanocomposites. Our goal was to determine their properties depending on the content of the nanofiller.
2) Of course, a summary of the injection process parameters can be presented in the appropriate table, however, the descriptive form is not defective, it also presents these parameters in a transparent way. However, recognizing the reasons of the reviewer that some of the items in the description of the injection moulding machine are generally available, we have significantly shortened the description.
3) According to the reviewer's suggestion, PE's mass contents have been changed in Table 1, so that they add up to 100%.
4) We could have used the statistical design of experiments, especially response surface methodology, for their mechancal and thermal analyses of the specimens followed by optimization, but the plan of the experiment adopted by us does not cover many variables within wide limits of their value variation. The fact that the answer surface method has not been used does not mean that there are no rules for good experimental practices, and that the results obtained are not unbiased. There are many examples of very good manuscripts published without a statistical design of experiments.
5) According to the reviewer's suggestion, the information on the number of replicates has been transferred from the captions under the drawings to the Methods section.
6) The reviewer mistakenly noticed that variations in the collected data point values presented as error bars are related to the different measurements made on the same processed material. Error bars is a standard deviation calculated from 10 measurements from different injections.
7) In the case of DSC studies, no statistical analysis is performed, which is confirmed by the results of DSC research published in:
a) Pagacz Joanna, Pielichowski Krzysztof, PVC/MMT nanocomposites: DSC with stochastic temperature modulation study at glass transition region, Journal of Thermal Analysis and Calorimetry 2013, 111 (2), 1571-1575.
b) Marta Worzakowska, Enelio Torres-Garcia, The effect of the grafting percentage of starch-g-poly(phenyl acrylate) copolymers on their pyrolysis and kinetics studied by the TG/DSC/FTIR/QMS , coupled method, Polymer Degradation and Stability 2017, 139,67-75.
and in many others.
8) Our experiment program assumed nanofiller changes in the range from 0 to 6% and was chosen on the basis of literature analysis. We found that this level is right, using higher values of nanofiller does not bring beneficial results, but only increases costs.
9) As we explained earlier, there was no reason to develop a statistical design of the experiment and its optimization. In many publications, the experiment program is presented in a similar way to ours and it is perfectly correct. Not in every study the results obtained must differ significantly so that the results do not overlap. It was written in the manuscript that in some cases the impact of a nanofiller is not important.
Once again, I would like to thank the Reviewer for the time devoted to reviewing the manuscript and for valuable observations.
Best regards,
Authors
Round 2
Reviewer 1 Report
In this work polyethylene composite materials filled with halloysite were studied. Please find below my comments and suggestions after this second revision.
Introduction: The introduction was modified, and the novelty of the work compared to previous research works was highlighted.
Keywords: Perhaps keywords are very general. I suggest including at least LDPE or halloysite
Page 3. Table 1. I suggest replacing “specimen” by “sample” as there may be more than one specimen of the same “sample”. Then, modify accordingly in the rest of the document, where necessary.
Page 6. DSC traces were included, as suggested and discussion of this part was modified accordingly.
After revising the document, once these issues are addressed, I consider it can be published at polymers.
Author Response
Dear Reviewer,
I would like to thank the Reviewer very much for his time and attention and for the effort he put into making this article more scientifically valuable. Keywords have been extended, as well as I changed the word "specimen" to "samples" and modified the text accordingly.
Best regards,
Authors
Reviewer 2 Report
Accepted
Author Response
Dear Reviewer,
I would like to thank the Reviewer very much for his time and attention and for the effort he put into making this article more scientifically valuable.
Best regards,
Authors,
Reviewer 3 Report
The authors addressed most of my concerns.
Author Response
Dear Reviewer,
I would like to thank the Reviewer very much for his time and attention and for the effort he put into making this article more scientifically valuable.
Best regards,
Authors